# GROND: A STEALTHY BACKDOOR ATTACK IN MODEL PARAMETER SPACE

## ABSTRACT

Recent research on backdoor attacks mainly focuses on invisible triggers in input space and inseparable backdoor representations in feature space to increase the backdoor stealthiness against defenses. We examine common backdoor attack practices that look at input-space or feature-space stealthiness and show that state-of-the-art stealthy input-space and feature-space backdoor attacks can be easily spotted by examining the parameter space of the backdoored model. Leveraging our observations on the behavior of the defenses in the parameter space, we propose a novel clean-label backdoor attack called `Grond`. We present extensive experiments showing that `Grond` outperforms state-of-the-art backdoor attacks on CIFAR-10, GTSRB, and a subset of ImageNet. Our attack limits the parameter changes through Adversarial Backdoor Injection, adaptively increasing the parameter-space stealthiness. Finally, we show how combining `Grond`'s Adversarial Backdoor Injection with commonly used attacks can consistently improve their effectiveness. Our code is available at `https://anonymous.4open.science/r/grond-557F`.

## 1 INTRODUCTION

While deep neural networks (DNNs) have achieved excellent performance on various tasks, they are known to be vulnerable to backdoor attacks. Backdoor attacks insert a secret functionality into a model that is activated when malicious inputs containing a specific trigger are provided to the model during inference. Backdoored DNNs can be crafted by training with poisoned data (Gu et al., 2019; Chen et al., 2017), controlling the training process (Bagdasaryan & Shmatikov, 2021; Nguyen & Tran, 2021; 2020), or directly modifying the weights of the victim model (Hong et al., 2022).

In early backdoor attacks (Gu et al., 2019; Chen et al., 2017; Liu et al., 2018b), triggers could induce noticeable changes that human inspectors or anomaly detectors can easily spot. To enhance *input space stealthiness*, smaller or more semantic-informative (to be mixed with clean samples) triggers are designed. However, most input-space stealthy backdoor attacks need to change labels of poisoned samples to the target class, i.e., dirty-label, which is also easy to detect (Chen et al., 2018). To this end, another line of backdoor attacks poisons the training data without changing the labels (Turner et al., 2019; Zeng et al., 2023), i.e., clean-label, which is more stealthy during poisoning but requires large trigger sizes. For example, (Zeng et al., 2023) uses $l_\infty$ norm of triggers to be upper bound by $16/255$ for backdoor training and $48/255$ at inference. In addition, despite the stealthiness concerning input images and labels, it has been widely observed that existing attacks introduce separability in the feature space, which can be exploited to develop backdoor defenses (Wang et al., 2022a; Xu et al., 2024b).

In response to feature-space defenses, state-of-the-art (SOTA) backdoor attacks focus on eliminating the separability in the feature space (Mo et al., 2024; Qi et al., 2023; Zhu et al., 2024; Shokri et al., 2020). However, our analysis shows that these attacks still require significant modifications to the model's parameters, and there is a lack of systematic evaluations against the latest *parameter-space* backdoor defenses. To this end, we evaluate ten attacks against nine parameter-space backdoor defenses, including four pruning-based and five fine-tuning-based defenses (See Section 4.2). Surprisingly, our experiments demonstrate that state-of-the-art backdoor attacks can be easily mitigated by parameter-space defenses, such as ANP (Wu & Wang, 2021), RNP (Li et al., 2023a), or FT-SAM (Zhu et al., 2023). More importantly, our systematic analysis reveals that even though some

Figure 1: Diagram illustrating the working mechanism of the Adversarial Backdoor Injection. On the left, Targeted Universarial Adversarial Perturbations (TUAP) are generated as backdoor patterns to be injected. In the middle, perturbed samples are iteratively used to train the model, and the model parameters are pruned to limit the magnitude of prominent backdoored weights. On the right, the output backdoored model that considers comprehensive stealthiness is deployed, where 1) the triggers are invisible, 2) the features of trigger samples are inseparable, and 3) the backdoored model weights are indistinguishable from benign model weights. Perturbations generated by TUAP are scaled up $10\times$ for visualization. Examples of poisoned images are provided in Appendix B.6.

backdoor attacks can resist several defenses, bypassing all defenses is non-trivial. For example, the Adap-blend attack (Qi et al., 2023), designed to eliminate the separability of latent features, can bypass most pruning-based defenses but cannot bypass fine-tuning-based defenses. These shortcomings could limit the application of backdoors in practice, such as DNN watermarking (Uchida et al., 2017).

To resolve these shortcomings, we propose a novel clean-label attack called Grond that considers *comprehensive stealthiness* to remain stealthy in the input, the feature, and the parameter space of the model. Grond achieves the input space stealthiness by using targeted universal adversarial perturbation (TUAP) (Moosavi-Dezfooli et al., 2017) as the trigger, where we only poison the target class to build a clean-label attack. To inject the backdoor, we propose a novel *Adversarial Backdoor Injection* mechanism that adaptively injects the backdoor during poisoning to achieve parameter space stealthiness. Specifically, we leverage the Lipschitz continuity of neuron activations to find backdoor-related suspicious and sensitive neurons in each poisoning epoch. Then, we conduct pruning on these neurons to eliminate the backdoor effect. As a result, after Adversarial Backdoor Injection, the backdoor is associated with neurons throughout the DNN rather than just focusing on a few prominent neurons, as illustrated in Figure 1.

We make the following contributions:

- We revisit state-of-the-art backdoor attacks regarding their stealthiness, showing that most attacks only aim for input space invisibility and feature space inseparability. Based on this finding, we examine the latest backdoor attacks and show that state-of-the-art stealthy input-space and feature-space backdoor attacks are vulnerable to parameter-space defenses.
- We propose a novel backdoor attack, Grond, that considers comprehensive stealthiness, taking input-, feature-, and parameter-space defenses into account. Extensive experiments demonstrate that Grond outperforms SOTA backdoor attacks against four pruning- and five fine-tuning-based defenses on CIFAR-10, GTSRB, and ImageNet200. We also show that Grond is resistant against five model detections and two input detections.
- We further verify our approach by binding Grond's Adversarial Backdoor Injection with other attacks. Experimental results demonstrate that our Adversarial Backdoor Injection could substantially improve the parameter space robustness of most backdoor attacks.

## 2 BACKGROUND & RELATED WORK

### 2.1 PRELIMINARIES ON BACKDOOR TRAINING

This paper considers a $C$-class classification problem with an $L$-layer CNN network $f = f_L \circ \cdots f_1$. Suppose that $\mathcal{D} = \{(\boldsymbol{x}_i, y_i)\}_{i=1}^N$ is the original training data, containing $N$ samples of $\boldsymbol{x}_i \in \mathbb{R}^{d_c \times d_h \times d_w}$ and its label $y \in \{1, 2, \ldots, C\}$. $d_c$, $d_h$, and $d_w$ are the number of input channels,

the height, and the width of the image, respectively. The attacker chooses a target class $t$ and creates a partially poisoned dataset $\mathcal{D}_p$ by poisoning generators $G_x$ and $G_y$, i.e., $\mathcal{D}_p = \mathcal{D}_c \cup \mathcal{D}_b$. $\mathcal{D}_c$ is the clean data from original dataset, $\mathcal{D}_b = \{(\boldsymbol{x}', y')|\boldsymbol{x}' = G_x(\boldsymbol{x}), y' = G_y(y), (\boldsymbol{x}, y) \in \mathcal{D} - \mathcal{D}_c\}$. In the clean-label setting, $G_y(y) = y$. For the dirty-label attacks, $G_y(y) = t$.

Our attack is a clean-label attack (Turner et al., 2019) following an all-to-one setting where the trigger should lead to a misclassification regardless of the original class of the poisoned sample. In the training stage, the backdoor is inserted into $f$ by minimizing the loss on $\mathcal{D}_p$:

$$\min_{\boldsymbol{\theta}} \mathcal{L}_{\mathcal{D}_p}(\boldsymbol{\theta}) = \mathop{\mathbb{E}}_{(\boldsymbol{x}, y) \in \mathcal{D}_p} \ell(f(\boldsymbol{x}; \boldsymbol{\theta}), y). \tag{1}$$

In the inference stage, the trained $f$ performs well on clean data $\hat{\boldsymbol{x}}$, but predicts $G_x(\hat{\boldsymbol{x}})$ as $G_y(\hat{y})$.

## 2.2 BACKDOOR ATTACKS

Backdoor attacks compromise the integrity of the victim model so that the model performs naturally on benign inputs but is misled to the target class by inputs containing the backdoor trigger. The trigger can be a visible pattern inserted into the model's input in **input space** or a property that affects the feature representation of the model's input in **feature space**. Eventually, however, the backdoored model's parameters in the **parameter space** will be altered regardless of the exact backdoor attack. To insert a backdoor, the attacker is assumed to control a small portion of the training data (Gu et al., 2019; Chen et al., 2017; Zhang et al., 2021) or also control the training process (Shokri et al., 2020; Nguyen & Tran, 2020; Bagdasaryan & Shmatikov, 2021; Nguyen & Tran, 2021; Wang et al., 2022b). Moreover, the backdoor can also be created by directly modifying the model's weights (Liu et al., 2017; Hong et al., 2022; Qi et al., 2022).

**Input-space attacks.** Traditional backdoor attacks typically use simple patterns as their triggers. For example, BadNets (Gu et al., 2019) uses a fixed patch, and Blend (Chen et al., 2017) mixes a Hello Kitty pattern into the images as the trigger. These non-stealthy triggers introduce abnormal data into training data and can be easily detected by human inspectors or defenses (Chen et al., 2018; Wang et al., 2019). To improve the stealthiness, various triggers are proposed to achieve *invisibility* in the input space. IAD (Nguyen & Tran, 2020) designed a dynamic solution in which the triggers vary among different inputs. WaNet (Nguyen & Tran, 2021) proposed the warping-based trigger, which is invisible to human inspection. Bpp (Wang et al., 2022b) used image quantization and dithering as the trigger, which makes imperceptible changes to images. Although these methods successfully build invisible triggers and bypass traditional defenses (Wang et al., 2019), they still introduce separable features and can be detected by feature-space defenses (Wang et al., 2022a; Xu et al., 2024b). These input-invisible attacks can be even more noticeable than input-visible attacks (BadNet, Blend) in the feature space (Xu et al., 2024a).

**Feature-space attacks.** Knowing the vulnerability of input-space attacks against feature-space defenses, backdoor attacks are improved for feature-space stealthiness. A common threat model of this attack type is assume additional control over the training process. For example, (Shokri et al., 2020; Zhao et al., 2022; Zhong et al., 2022) directly designed loss functions to minimize the difference between the backdoor and benign features. Aside from design loss penalties, TACT (Tang et al., 2021) and SSDT (Mo et al., 2024) point out that source-specific (poison only the specified source classes) attack helps obscuring the difference in features between benign and backdoor samples. In addition, (Qi et al., 2023) proposed Adap-blend and Adap-patch, which obscures benign and backdoor features by 1) including poisoned samples with the correct label, 2) asymmetric triggers (using a stronger trigger at inference time), and 3) trigger diversification (using diverse variants of the trigger during training). Unfortunately, existing attacks lack systematic evaluation against the latest defenses. For example, Adap-blend can be thoroughly mitigated by recent works (Zhu et al., 2023; Xu et al., 2024b;a).

## 2.3 BACKDOOR DEFENSES

Backdoor defenses can be classified into detection and mitigation. Detection refers to determining whether a model is backdoored (*model detection*) (Wang et al., 2019; Liu et al., 2019; Zhao et al., 2022; Wang et al., 2023; Xu et al., 2024b) or a given input is applied with a trigger (*input detection*) (Gao et al., 2019; Guo et al., 2023; Mo et al., 2024). Model detection by trigger inversion is

considered one of the most general defenses against backdoors (Wang et al., 2022a; 2023; Xu et al., 2024b; Zhu et al., 2024). The inversed trigger could determine whether the model is backdoored and be used for backdoor unlearning. For example, NC (Wang et al., 2019) inverses input space triggers and determines the backdoor by selecting abnormally smaller triggers.

Mitigation refers to erasing the backdoor effect from the victim model by pruning the backdoor-related neurons (*pruning-based* defenses) (Liu et al., 2018a; Wu & Wang, 2021; Zheng et al., 2022; Li et al., 2023a) or unlearning the backdoor trigger (*fine-tuning-based* defenses) (Zhu et al., 2023; Zeng et al., 2022; Min et al., 2023; Xu et al., 2024b). These methods attempt to remove the neurons associated with backdoors. For example, ANP (Wu & Wang, 2021) prunes neurons that are more sensitive to adversarial neuron noise, and FT-SAM (Zhu et al., 2023) combines sharpness-aware minimization with fine-tuning to decrease the norms of backdoor neurons.

## 3 METHODOLOGY FOR COMPREHENSIVE STEALTHINESS

### 3.1 THREAT MODEL

**Attacker's goal.** The attacker provides pre-trained models to users. The aim is to inject backdoors into the pre-trained model so that the model performs well on clean inputs but predicts the attacker-chosen target label when receiving inputs with a backdoor trigger, i.e., an all-to-one attack.

**Attacker's knowledge.** The attacker has white-box access to the training processes, the training data, and the model weights. During training, poisoned images do not contain visible patterns for human inspectors, so the labels of poisoned images are the same as the images' original class, i.e., clean-label. During inference, the backdoor trigger is invisible to human inspectors.

**Attacker's capabilities.** The attacker can train a well-performed surrogate model to generate TUAP, which is used by the attacker to perturb the victim model's input. Additionally, the attacker can alter the model's weights during training. Table 7 in Appendix A.2 shows that the threat model of `Grond` is aligned with baseline attacks.

### 3.2 BACKDOOR ATTACKS MUST CONSIDER PARAMETER SPACE DEFENSES

Early backdoor attacks that introduce noticeable changes in either input (Gu et al., 2019; Chen et al., 2017) or feature space (Nguyen & Tran, 2021; 2020) have been empirically shown powerful, even with very low poisoning rates (Gu et al., 2019; Zeng et al., 2023). Focusing on the backdoor-introduced noticeable changes, backdoor defenses are improved to distinguish backdoor patterns in either input or feature space (Wang et al., 2022a; Lin et al., 2024). Meanwhile, backdoor attacks are optimized to increase stealthiness in input (Nguyen & Tran, 2021) or feature space (Qi et al., 2023). However, we point out that, regardless of the implementations in the input of feature spaces, the backdoor behaviors are eventually embedded and reflected in the backdoored model parameters (Liu et al., 2019; Wu & Wang, 2021; Li et al., 2023a). As seen in Table 1, our experiments demonstrate that existing attacks (including both input- and feature-space) are not robust against parameter-space backdoor defenses, such as ANP (Wu & Wang, 2021) (pruning backdoor-related parameters) and FT-SAM (Zhu et al., 2023) (shrinking the norms of backdoor-related neurons). This finding indicates that stronger backdoor attacks should consider the parameter-space defenses, besides input- and feature-space defenses, to achieve *comprehensive stealthiness*.

### 3.3 GROND FOR COMPREHENSIVE STEALTHINESS

We propose a stealthy backdoor attack, `Grond`, that considers comprehensive stealthiness, i.e., stealthiness in input, feature, and parameter space. `Grond` includes two key parts: Backdoor generation and Adversarial Backdoor Injection.

**Backdoor generation for input-space stealthiness.** We use imperceptible adversarial perturbations to generate *invisible* backdoor triggers, inspired by adversarial example studies (Moosavi-Dezfooli et al., 2017; Zhang et al., 2021). Specifically, we use Targeted Universal Adversarial Perturbations (TUAP) as the imperceptible perturbations. TUAP contains non-robust but generalizable semantic information (Tsipras et al., 2019), which correlates with the benign functions of the victim model and

Table 1: A summary (supported by our experiments) of existing attacks and defenses. ● indicates that the backdoor attack can bypass the defense with an Attack Success Rate (ASR) higher than 60%. ○ indicates that the backdoor attack fails to bypass the defense. The 60% level is selected based on the average behaviors seen in Tables 2 and 3.

| Type | Attack | Pruning-Based | | | | Fine-Tuning-Based | | | | |
|------|--------|----|-----|-----|-----|-----------|--------|-------|-----|-----------|
| | | FP | ANP | CLP | RNP | vanilla FT | FT-SAM | I-BAU | FST | BTI-DBF(U) |
| Dirty-Label | BadNets | ● | ○ | ○ | ○ | ○ | ○ | ● | ○ | ○ |
| | Blend | ● | ○ | ○ | ○ | ● | ○ | ○ | ● | ● |
| | WaNet | ○ | ○ | ○ | ○ | ○ | ○ | ○ | ○ | ○ |
| | IAD | ○ | ○ | ○ | ○ | ○ | ○ | ○ | ○ | ○ |
| | AdvDoor | ● | ○ | ○ | ○ | ● | ○ | ○ | ● | ● |
| | Bpp | ○ | ○ | ○ | ○ | ○ | ○ | ○ | ○ | ○ |
| | SSDT | ○ | ○ | ○ | ○ | ○ | ○ | ○ | ○ | ○ |
| | Adap-blend | ● | ● | ● | ○ | ● | ○ | ○ | ○ | ○ |
| Clean-Label | LC | ● | ○ | ○ | ○ | ● | ○ | ○ | ○ | ○ |
| | Narcissus | ● | ○ | ● | ● | ● | ○ | ○ | ○ | ● |
| | Grond | ● | ● | ● | ● | ● | ● | ● | ● | ● |

shortens the distance between poisoned data and the target classification region (Zhang et al., 2021). Consequently, backdoor patterns tend to make fewer prominent changes to the victim network.

Similar to (Zeng et al., 2023; Zhang et al., 2021), TUAP is generated on a well-trained surrogate model that is trained on the clean training set. The architecture and parameters of the surrogate model do not necessarily need to be the same as the victim model. Formally, TUAP is optimized following the PGD (Madry et al., 2018) algorithm to decrease the surrogate model's cross-entropy loss that takes as inputs the adversarial examples (the poisoned samples in our case) and the target class samples. This procedure is described formally as follows:

$$\min_{\boldsymbol{\delta} \in S} \mathcal{L}_{\mathcal{D}}(\boldsymbol{\theta}) = \mathbb{E}_{(\boldsymbol{x},y) \in \mathcal{D}} \ell(f(\boldsymbol{x} + \boldsymbol{\delta}; \boldsymbol{\theta}), t),$$
$$S = B(\boldsymbol{\delta}; \epsilon) = \{\boldsymbol{\delta} \in \mathbb{R}^{d_c \times d_h \times d_w} : ||\boldsymbol{\delta}||_{\infty} \leq \epsilon\}, \tag{2}$$

where $\boldsymbol{\delta}$ is the generated TUAP that will be used as a backdoor trigger; thus, $G_x(\boldsymbol{x}) = \boldsymbol{x} + \boldsymbol{\delta}$. $S$ is the ball function with the radius $\epsilon$, and the small $\epsilon$ guarantees the invisibility of the backdoor trigger as it controls the perturbation's magnitude.

The backdoor is injected during training by poisoning some training data from the target class, i.e., applying the TUAP to the training data. In the inference stage, our backdoor is activated by the same trigger. Please note that we do not scale up the inference-stage trigger size ($\epsilon$). This is a more strict condition than present in previous clean-label attacks (Turner et al., 2019; Zeng et al., 2023). For instance, Narcissus (Zeng et al., 2023) poisons the training data through a trigger with $\epsilon = 16$ and uses a larger size of trigger ($\epsilon = 48$) at the inference stage. The motivation for our small-size trigger ($\epsilon = 8$) is invisibility.

**Adversarial Backdoor Injection for parameter-space stealthiness.** Backdoor neurons (i.e., trigger-related neurons) regularly show higher activation values for inputs that contain the trigger, which results in powerful performance (Liu et al., 2019; Wang et al., 2022a; Lin et al., 2024). To this end, backdoor training needs to substantially increase the magnitude of parameters of backdoor neurons (Wu & Wang, 2021; Li et al., 2023a; Zheng et al., 2022), which harms the parameter-space stealthiness of backdoor attacks.

One way to find the sensitive neurons with higher activation values is to analyze the Lipschitz continuity of the network. Leveraging this fact, we introduce a novel backdoor training mechanism, *Adversarial Backdoor Injection*, to increase the parameter-space backdoor stealthiness. Specifically, each neuron's Upper bound of Channel Lipschitz Condition (UCLC (Zheng et al., 2022)) is calculated, based on which the weights of these suspicious neurons are set to the mean of all neurons' weights in the corresponding layer after every training epoch. In our implementation, we use the weights before every batch normalization as the neuron weights, which corresponds to the channel setting in UCLC. We prune neurons by substituting their weights with the mean ones because pruning to zeros makes the training unable to converge. Formally, the $k_{th}$ parameter of the $l_{th}$ layer,

$\boldsymbol{\theta}_l^{(k)}$, is updated as follows where $u = 3$ is a fixed threshold,

$$\boldsymbol{\theta}_l^{(k)} := \begin{cases} \text{mean}(\boldsymbol{\theta}_l), & \sigma(\boldsymbol{\theta}_l^{(k)}) > \text{mean}(\sigma(\boldsymbol{\theta}_l)) + u \times \text{std}(\sigma(\boldsymbol{\theta}_l)) \\ \boldsymbol{\theta}_l^{(k)}, & \text{otherwise}, \end{cases} \quad (3)$$

and $\sigma$ is the UCLC value of the given weights. The measure for quantifying backdoor relevance can be changed from UCLC to others, such as the distance of neuron outputs when receiving benign and backdoor inputs, where a larger distance means the neuron is more relevant to backdoor behaviors and can be pruned. We use the modified UCLC for training efficiency as UCLC is data-free, which does not require calculation based on the outputs of neurons.

In adversarial training (Madry et al., 2018), adversarial examples are introduced during training to increase the model's robustness during inference. Similarly, during the Adversarial Backdoor Injection, we use backdoor defenses to increase the resistance of backdoor attacks to parameter-space defenses. At the end of each training epoch, Adversarial Backdoor Injection prunes the trained model to decrease the weights of backdoor neurons. Iteratively, backdoored neurons spread across the whole model instead of forming a few prominent backdoor neurons, as illustrated in Figure 1.

**About feature-space stealthiness.** We hypothesize that feature-space stealthiness is a by-product of parameter-space and input-space stealthiness since the variation of feature maps is strongly correlated with model parameters and inputs. Figure 2 shows that `Grond` can substantially increase the feature-space stealthiness. More details can be found in Equation 4 in Section 4.2.

## 4 EXPERIMENTAL EVALUATION

### 4.1 EXPERIMENTAL SETUP

**Datasets and Architectures.** We follow the common settings in existing backdoor attacks and defenses and conduct experiments on CIFAR-10 (Krizhevsky et al., 2009), GTSRB (Stallkamp et al., 2012), and a subset of ImageNet (Deng et al., 2009) with 200 classes (ImageNet200). More details about the datasets can be found in Appendix A.1. The primary evaluation is performed using ResNet18 (He et al., 2016). Moreover, we evaluate `Grond` using four architectures, three common ones, VGG16 (Simonyan & Zisserman, 2015), DenseNet121 (Huang et al., 2017), EfficientNet-B0 (Tan, 2019), and one recent architecture InceptionNeXt (Yu et al., 2024). Due to space limits, the results on different architectures and different surrogate models are given in Appendix B.3.

**Attack Baselines.** `Grond` is compared with ten representative attacks: BadNets (Gu et al., 2019), Blend (Chen et al., 2017), WaNet (Nguyen & Tran, 2021), IAD (Nguyen & Tran, 2020), Adv-Door (Zhang et al., 2021), BppAttack (Wang et al., 2022b), LC (Turner et al., 2019), Narcissus (Zeng et al., 2023), Adap-Blend (Qi et al., 2023), and SSDT (Mo et al., 2024). The poisoning rate is 5% for all attacks. `Grond` is evaluated using multiple poisoning rates to provide a more complete analysis of its behavior. Backdoor attack implementation details can be found in Appendix A.3.

**Defense Baselines.** We evaluate `Grond` and baseline attacks with 16 representative defenses, including **four pruning-based** methods (FP (Liu et al., 2018a), ANP (Wu & Wang, 2021), CLP (Zheng et al., 2022), and RNP (Li et al., 2023a)), **five fine-tuning-based** methods (vanilla FT, FT-SAM (Zhu et al., 2023), I-BAU (Zeng et al., 2022), FST (Min et al., 2023), and BTI-DBF(U) (Xu et al., 2024b)), **five backdoor model detections** (NC (Wang et al., 2019), Tabor (Guo et al., 2020), FeatureRE (Wang et al., 2022a), Unicorn (Wang et al., 2023), and BTI-DBF (Xu et al., 2024b)), and **two backdoor input detections** (Scale-up (Guo et al., 2023) and IBD-PSC (Hou et al., 2024)). Backdoor defense details of hyperparameters can be found in Appendix A.4.

### 4.2 MAIN RESULTS

Overall, `Grond` performs better than all baseline attacks. `Grond` achieves 7.18% higher ASR on average than the best baseline attack, Narcissus, against four pruning-based mitigations. The five fine-tuning mitigations show more powerful defense capability, and `Grond` achieves 29.25% higher ASR on average than Narcissus.

**Pruning-tuning-based mitigations.** Table 2 shows the results of all attacks against four pruning-based defenses. Backdoor pruning assumes the separability of benign and backdoor neurons, and

Table 2: Pruning-based mitigations against backdoored ResNet18 on CIFAR-10. BA refers to benign accuracy on clean data, ASR to attack success rate, and pr to the poisoning rate.

| Attack | No Defense | | FP | | ANP | | CLP | | RNP | | Average | |
|---|---|---|---|---|---|---|---|---|---|---|---|---|
| | BA | ASR | BA | ASR | BA | ASR | BA | ASR | BA | ASR | BA | ASR |
| BadNets | 93.13 | 100 | 92.42 | 71.71 | 91.60 | 1.06 | 88.99 | 49.02 | 84.04 | 13.82 | 89.26 | 33.90 |
| Blend | 94.42 | 100 | 93.08 | 99.99 | 93.57 | 0.33 | 90.30 | 0.54 | 94.63 | 57.98 | 92.89 | 39.71 |
| WaNet | 93.60 | 99.37 | 92.96 | 4.60 | 91.08 | 0.49 | 91.53 | 2.12 | 92.86 | 3.17 | 92.11 | 2.59 |
| IAD | 92.88 | 97.10 | 91.96 | 1.22 | 92.84 | 0.71 | 92.24 | 0.74 | 92.72 | 0.42 | 92.44 | 0.77 |
| AdvDoor | 93.97 | 100 | 93.37 | 98.69 | 91.46 | 28.83 | 89.22 | 6.13 | 90.17 | 44.60 | 91.05 | 44.56 |
| Bpp | 94.19 | 99.93 | 93.38 | 18.89 | 92.96 | 2.97 | 93.37 | 1.89 | 92.20 | 5.79 | 92.98 | 7.39 |
| LC | 94.31 | 100 | 92.22 | 93.57 | 91.02 | 24.43 | 90.96 | 0.38 | 82.70 | 33.60 | 89.23 | 37.99 |
| Narcissus | 93.58 | 99.64 | 93.49 | 96.54 | 89.76 | 49.18 | 93.19 | 97.82 | 91.10 | 94.59 | 91.88 | 84.53 |
| SSDT | 93.70 | 90.30 | 93.41 | 0.80 | 93.88 | 0.60 | 93.66 | 1.20 | 93.99 | 3.30 | 93.74 | 1.47 |
| Adap-blend | 92.74 | 99.67 | 92.06 | 95.50 | 86.48 | 67.73 | 92.49 | 99.62 | 78.63 | 1.56 | 87.42 | 66.10 |
| Grond (pr=5%) | 93.43 | 98.04 | 93.09 | 99.73 | 91.43 | 94.01 | 93.29 | 87.89 | 91.83 | 85.22 | 92.41 | 91.71 |
| Grond (pr=1%) | 94.26 | 93.51 | 93.31 | 96.32 | 92.94 | 91.48 | 94.33 | 87.56 | 92.13 | 94.87 | 93.18 | 92.56 |
| Grond (pr=0.5%) | 94.52 | 87.06 | 93.32 | 90.96 | 93.87 | 84.04 | 94.52 | 86.82 | 91.99 | 84.63 | 93.43 | 86.61 |

greater weights accompany backdoor neurons. Then, the backdoor can be removed by pruning the backdoor-related neurons, i.e., reducing their weights to zero. We observe that input-space attacks, BadNets and Blend, perform better than input-space stealthy attacks, e.g., WaNet and Bpp, because input-space stealthy attacks introduce significant separability in the feature space (see Figure 2).

Table 3: Fine-tuning-based mitigations against backdoored ResNet18 on CIFAR-10.

| Attack | vanilla FT | | FT-SAM | | I-BAU | | FST | | BTI-DBF(U) | | Average | |
|---|---|---|---|---|---|---|---|---|---|---|---|---|
| | BA | ASR | BA | ASR | BA | ASR | BA | ASR | BA | ASR | BA | ASR |
| BadNets | 91.07 | 43.96 | 92.01 | 2.84 | 90.87 | 97.48 | 92.40 | 13.10 | 91.26 | 13.12 | 91.52 | 34.10 |
| Blend | 91.64 | 99.61 | 92.52 | 1.73 | 91.84 | 8.84 | 93.40 | 100 | 91.86 | 100 | 92.25 | 62.04 |
| WaNet | 91.11 | 0.99 | 90.89 | 1.03 | 87.98 | 0.81 | 92.17 | 0.04 | 90.30 | 4.89 | 90.49 | 1.55 |
| IAD | 90.83 | 2.16 | 92.18 | 2.87 | 88.40 | 15.68 | 91.29 | 0.00 | 89.54 | 1.59 | 90.45 | 4.46 |
| AdvDoor | 91.25 | 68.68 | 92.18 | 1.23 | 89.29 | 16.99 | 91.06 | 99.99 | 90.25 | 100 | 90.81 | 57.38 |
| Bpp | 91.36 | 3.40 | 91.38 | 1.00 | 92.06 | 6.46 | 93.23 | 26.83 | 90.61 | 2.73 | 91.73 | 8.08 |
| LC | 90.26 | 88.52 | 91.46 | 1.91 | 85.87 | 5.11 | 91.80 | 13.11 | 90.71 | 4.37 | 90.02 | 22.60 |
| Narcissus | 91.70 | 92.91 | 91.76 | 23.98 | 91.48 | 51.74 | 90.06 | 54.22 | 90.94 | 98.11 | 91.19 | 64.19 |
| SSDT | 93.74 | 0.70 | 93.15 | 0.60 | 90.27 | 3.10 | 92.85 | 0.20 | 90.79 | 1.40 | 92.16 | 1.20 |
| Adap-blend | 92.42 | 98.73 | 91.23 | 22.40 | 85.38 | 37.31 | 90.91 | 1.19 | 89.17 | 7.09 | 89.82 | 33.34 |
| Grond (pr=5%) | 91.75 | 94.28 | 92.02 | 80.07 | 90.39 | 93.92 | 93.27 | 99.92 | 91.88 | 99.00 | 91.86 | 93.44 |
| Grond (pr=1%) | 91.41 | 85.52 | 92.83 | 79.17 | 87.89 | 91.34 | 93.21 | 96.59 | 90.66 | 88.69 | 91.20 | 88.26 |
| Grond (pr=0.5%) | 91.42 | 82.96 | 92.34 | 76.92 | 89.83 | 79.68 | 93.44 | 92.71 | 90.39 | 91.83 | 91.48 | 84.82 |

**Fine-tuning-based mitigations.** Table 3 shows the backdoor performance against five fine-tuning-based defenses. The fine-tuning defense results again demonstrate that attacks with prominent backdoor feature loss are easily mitigated. Additionally, experimental results show that fine-tuning-based defenses could outperform pruning-based defenses. For example, Narcissus and Adap-Blend can achieve ASRs higher than 60% against three out of four pruning-based defenses but are much less effective against most fine-tuning-based methods. FT-SAM performs the best among all the defense baselines in Tables 2 and 3, compromising the effectiveness of all attack baselines. One important reason is that FT-SAM adopts Sharpness-Aware Minimization (Foret et al., 2021) to adjust the outlier of weight norm (large norms) to remove the potential backdoor. The large outlier is introduced by existing attacks to guarantee a high ASR (Liu et al., 2019), which also causes large differences when receiving benign and backdoor inputs (see Figure 4). However, Grond can bypass FT-SAM since it deliberately decreases the weights of backdoor neurons, compromising the core working mechanism of FT-SAM.

**Backdoor analysis by feature space inversion.** We provide a feature space analysis on different attacks following BTI-DBF (Xu et al., 2024b) and BAN (Xu et al., 2024a). They assume the correct prediction, i.e., low loss, is made only on benign features, so the features related to backdoor prediction lead to high loss. In particular, the benign and inversed backdoor features (which lead to high loss) are disentangled as follows:

$$\min_{\boldsymbol{m}} \sum_{(\boldsymbol{x},y)\in\mathcal{D}_l} \Big[ \mathcal{L}\big(f_L \circ (g(\boldsymbol{x}) \odot \boldsymbol{m}), y\big) - \mathcal{L}\big(f_L \circ (g(\boldsymbol{x}) \odot (1-\boldsymbol{m}), y)\big) + \lambda|\boldsymbol{m}| \Big], \qquad (4)$$

where $g = f_{L-1} \circ \cdots f_1$, i.e., $g$ is $f$ without the classify head. $\boldsymbol{m}$ is the mask for the latent features. $\mathcal{D}_l$ is a set of a few local benign samples with correct labels. Details can be found in Appendix A.5.

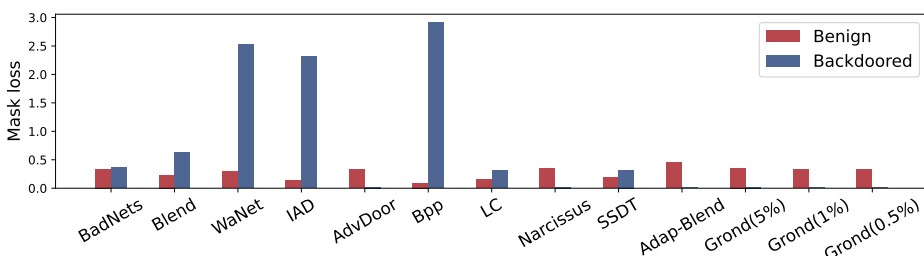

Figure 2: Benign and inversed backdoor feature loss (by Equation 4) for all baseline attacks. Large backdoored loss indicates that the backdoor is prominent in the feature space.

Figure 2 demonstrates benign (the first term in Equation 4) and inversed backdoor (the second term in Equation 4) feature loss for attack baselines. Assuming Equation 4 can disentangle prominent backdoor features, attacks that generate prominent inversed backdoor features can be easily mitigated by both pruning-based and fine-tuning-based defenses. Notice in Figure 2 that the inversed backdoor feature losses of WaNet, IAD, and Bpp are much more prominent (higher) than others. Defenses are more effective against WaNet, IAD, and Bpp than other attacks due to the prominent features, as shown in Tables 2 and 3. In contrast, backdoor attacks with lower inversed backdoor feature loss values, AdvDoor, Narcissus, Adap-blend, and `Grond`, are less affected by defenses.

Table 4: The detection results using ResNet18 and CIFAR-10. Bd. refers to the number of models determined as backdoor models. Acc. refers to the detection accuracy.

| Poisoning Rate | NC | | Tabor | | FeatureRE | | Unicorn | | BTI-DBF | |
|---|---|---|---|---|---|---|---|---|---|---|
| | Bd. | Acc. | Bd. | Acc. | Bd. | Acc. | Bd. | Acc. | Bd. | Acc. |
| 5% | 5 | 25% | 5 | 25% | 0 | 0% | 0 | 0% | 3 | 15% |
| 1% | 2 | 10% | 2 | 10% | 0 | 0% | 0 | 0% | 5 | 25% |
| 0.5% | 1 | 5% | 0 | 0% | 0 | 0% | 0 | 0% | 3 | 15% |

**Backdoor detection.** Following previous works (Xu et al., 2024b;a), we choose five representative backdoor model detections for evaluation. The model detection refers to determining whether a given model is backdoored. We train 20 models for each poisoning rate with different random seeds. Then, we report the number of models being detected as backdoor models. Table 4 shows all detections fall short in detecting `Grond`, where NC, Tabor, and BTI-DBF can find a small part of backdoor models, while FeatureRE and Unicorn cannot detect any of them. For featureRE, we conjecture that it is over-dependent on the separability in the feature space, but `Grond` does not rely on prominent backdoor features according to Figure 2. For Unicorn, the false positive rate is high, and it tends to report every class as the backdoor target, even on models trained with benign data only. We also show that `Grond`-generated backdoor samples can resist input detection that determines whether or not a given input includes a backdoor trigger in Table 10 in Appendix B.1.

**On ImageNet200 and GTSRB.** Real-world classification tasks may involve more classes than ten, such as GTSRB (43 classes) and ImageNet200 (200 classes), where the percentage of each class in the dataset will be much less than 10%. We target InceptionNext-Small on Imagenet200 and ResNet18 on GTSRB. The $l_\infty$ norm perturbation budget of TUAP is $\epsilon = 16$ for GTSRB and $\epsilon = 8$ for ImageNet200 for invisible perturbations. Table 5 demonstrates that `Grond` is still effective on datasets with more classes and higher resolutions, especially against the most powerful parameter-space defense, FT-SAM.

### 4.3 ABLATION STUDY

There are two components in `Grond`: the TUAP trigger and Adversarial Backdoor Injection. We conduct an ablation study on two architectures with CIFAR10 to demonstrate the effect of each component. Specifically, the ablation is designed by removing the Adversarial Backdoor Injection. Table 6 shows the ablation results. Without Adversarial Backdoor Injection, CLP or FT-SAM can defend against the clean-label attack with the TUAP trigger. These results verify the effectiveness of the Adversarial Backdoor Injection.

Table 5: Backdoor performance of `Grond` and baseline attacks on ImageNet200 and GTSRB.

| Datasets | Attack | No Defense | | FT-SAM | | I-BAU | | CLP | | Average | |
|---|---|---|---|---|---|---|---|---|---|---|---|
| | | BA | ASR | BA | ASR | BA | ASR | BA | ASR | BA | ASR |
| ImageNet200 | BadNets | 80.65 | 91.03 | 79.89 | 2.21 | 70.28 | 26.06 | 70.74 | 64.86 | 73.64 | 31.04 |
| | Blend | 80.70 | 95.63 | 80.19 | 0.39 | 76.13 | 30.81 | 80.02 | 23.38 | 78.78 | 18.19 |
| | WaNet | 81.24 | 99.97 | 80.41 | 0.66 | 75.67 | 47.27 | 77.18 | 99.78 | 77.75 | 49.24 |
| | IAD | 79.74 | 99.98 | 75.49 | 0.68 | 77.44 | 15.18 | 76.97 | 84.49 | 76.63 | 33.45 |
| | AdvDoor | 80.72 | 100 | 79.52 | 98.90 | 74.03 | 61.31 | 77.90 | 100 | 77.15 | 86.74 |
| | Bpp | 81.36 | 92.74 | 79.37 | 1.05 | 76.53 | 3.21 | 80.10 | 2.34 | 78.67 | 2.19 |
| | Narcissus | 81.73 | 81.28 | 80.00 | 83.37 | 77.03 | 56.19 | 80.99 | 86.37 | 79.34 | 75.31 |
| | SSDT | 75.45 | 100 | 78.19 | 76.00 | 76.26 | 22.00 | 76.02 | 94.00 | 76.82 | 64.00 |
| | Grond | 80.92 | 94.11 | 79.05 | 95.05 | 76.89 | 87.75 | 80.29 | 93.83 | 78.74 | 92.21 |
| GTSRB | BadNets | 97.19 | 100 | 95.57 | 0.48 | 92.02 | 29.22 | 96.38 | 0.47 | 94.66 | 10.06 |
| | Blend | 95.92 | 100 | 93.36 | 0.21 | 92.64 | 38.27 | 93.21 | 0.00 | 93.07 | 12.83 |
| | WaNet | 98.69 | 99.77 | 92.18 | 0.45 | 91.25 | 0.00 | 90.14 | 18.14 | 91.19 | 6.19 |
| | IAD | 99.08 | 99.65 | 92.72 | 0.10 | 90.11 | 0.35 | 98.08 | 14.63 | 93.64 | 5.03 |
| | AdvDoor | 95.80 | 99.99 | 93.94 | 32.26 | 92.67 | 38.20 | 90.09 | 66.39 | 92.23 | 45.62 |
| | Bpp | 98.69 | 99.93 | 91.27 | 0.00 | 92.61 | 0.23 | 97.16 | 2.29 | 93.68 | 0.84 |
| | Narcissus | 95.60 | 97.18 | 93.61 | 54.55 | 92.87 | 80.74 | 93.99 | 97.60 | 93.49 | 77.63 |
| | SSDT | 96.02 | 77.78 | 93.11 | 0.00 | 90.82 | 0.00 | 94.65 | 19.31 | 92.86 | 6.44 |
| | Grond | 95.83 | 95.36 | 93.80 | 71.84 | 93.13 | 94.30 | 91.28 | 93.19 | 92.74 | 86.44 |

Table 6: Ablation study of `Grond`.

| Victim | Method | No Defense | | CLP | | FT-SAM | |
|---|---|---|---|---|---|---|---|
| | | BA | ASR | BA | ASR | BA | ASR |
| ResNet18 | TUAP Trigger | 93.86 | 98.61 | 91.15 | 3.97 | 91.80 | 51.77 |
| | +Adversarial Injection | 93.43 | 98.04 | 93.29 | 87.89 | 92.02 | 80.07 |
| InceptionNeXt-Tiny | TUAP Trigger | 87.81 | 96.81 | 87.72 | 96.57 | 87.06 | 2.37 |
| | +Adversarial Injection | 87.06 | 96.86 | 86.93 | 96.87 | 86.50 | 92.02 |

## 4.4 ADVERSARIAL BACKDOOR INJECTION IMPROVES OTHER BACKDOOR ATTACKS

To evaluate the generalizability of `Grond`, we combine our Adversarial Backdoor Injection with all baseline attacks in order to improve their resistance against parameter-space defenses. Figure 3 demonstrates that Adversarial Backdoor Injection is effective for all attacks when evaluating against the parameter-space defense ANP, where ASRs increase after adversarial injection, especially for BadNets, Blend, AdvDoor, Narcissus, and Adap-Blend. The improvement for feature space attacks (WaNet, IAD, and Bpp) is incremental. We conjecture that feature space attacks rely too much on prominent features as their modification in the input space is minor. To activate the backdoor with such minor input modifications, the prominent features are required in the feature space. In addition, Figure 5 in Appendix B.2 shows the results of Adversarial Backdoor Injection without defense, demonstrating that it does not harm in general the BA and ASR when no defense is applied.

## 4.5 ORACLE BACKDOOR ANALYSIS BY TAC VALUES

This section provides an oracle experimental analysis utilizing the trigger information. Specifically, we use the TAC values (Zheng et al., 2022) to quantify the relevance of a neuron to the backdoor behavior according to its output when receiving benign and backdoor inputs. A higher TAC value indicates that the neuron is strongly relevant for backdoor behaviors. Based on the TAC analysis, we can prune the neurons with high TAC values in the backdoored model. TAC-based pruning is powerful as it directly uses the trigger information. However, TAC analysis cannot be used as a defense in practice because the trigger information is not accessible to the defender. Thus, we only use TAC to provide backdoor analysis. TAC details can be found in Appendix B.4.

Figure 4 shows the pruning results of `Grond` and three baseline attacks (TAC plots for other attacks are in Appendix B.5). The first row of Figure 4 provides the pruning results. The second row contains the TAC values plots of neurons in the $4_{th}$ layer (the layer before the classification head) of ResNet18. It is clear that the backdoor and benign behaviors of baseline attacks can be disentangled by pruning neurons with high TAC values. However, for `Grond`, pruning neurons with high

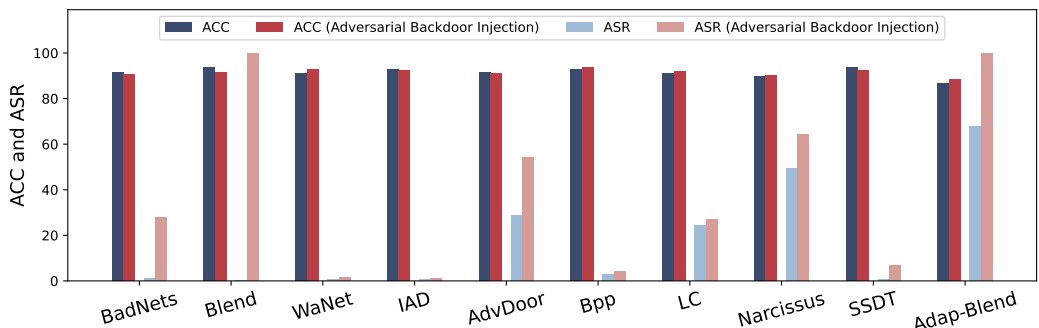

Figure 3: Benign accuracy and ASR of backdoor attacks before and after Adversarial Backdoor Injection against parameter-space defense ANP.

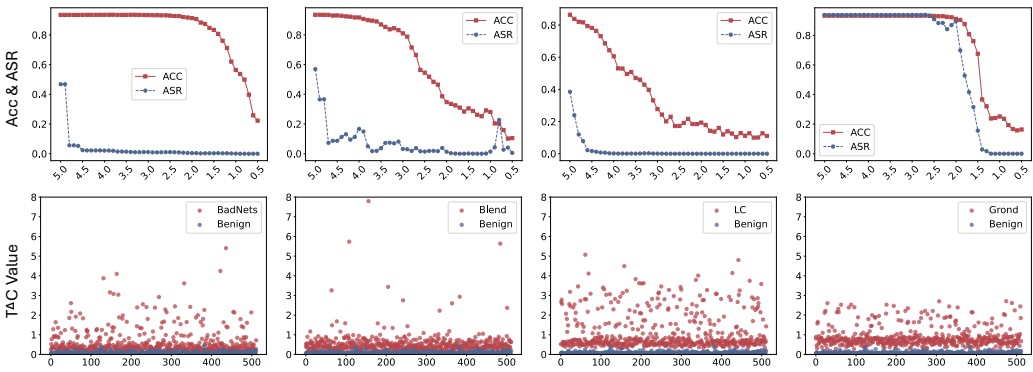

Figure 4: The TAC (trigger activated changes) (Zheng et al., 2022) plot demonstrates that other attacks inject the backdoor to a few prominent neurons, while `Grond`'s neurons are more compact. Higher TAC values represent a stronger relation between corresponding neurons and the backdoor effect. The first row shows the performance of pruning neurons with high TAC values. The second row provides the TAC values for corresponding neurons. Please refer to Appendix B.4 for more details on TAC, B.5 for plots for other attacks, and Figure 7 for sorted TAC values.

TAC values will also decrease benign accuracy, which means the backdoor neurons are not easily distinguishable from benign neurons. The analysis supports our statement that `Grond` spreads the backdoor to more neurons instead of a few prominent ones.

## 5 CONCLUSIONS AND LIMITATIONS

This paper examines input-space and feature-space backdoor attacks in the parameter space, showing that state-of-the-art backdoor attacks are surprisingly vulnerable to parameter-space defenses. To overcome this shortcoming, we propose a novel clean-label backdoor attack, `Grond`, that considers comprehensive stealthiness, including input-, feature-, and parameter-space stealthiness. `Grond` achieves state-of-the-art performance by leveraging adversarial examples and adaptively limiting the backdoored model's parameter changes during the backdoor injection to improve the backdoor stealthiness. We also show that `Grond`'s Adversarial Backdoor Injection can consistently improve other backdoor attacks against parameter space defenses.

While current backdoor defenses are ineffective against `Grond`, we anticipate defenses considering the design of `Grond` could mitigate `Grond` in the future. Moreover, we consider only one adversarial perturbation method, TUAP, and one Adversarial Backdoor Injection method, modified UCLC. As we pointed out in the oracle analysis in Section 4.5, different adversarial injection and adversarial perturbation methods are also promising under the `Grond` framework. We leave the exploration to future work.

ETHICS STATEMENT

Our adversarial experiments are conducted only in the laboratory environment, which has no effect in the realistic environment. Although existing defenses may not be able to mitigate our attack, our work encourages researchers to design defenses based on the root cause of the backdoor attack rather than defeat current attacks. We also believe our work induces positive impacts on other related fields, such as using our attack as a model watermarking technology for intellectual property protection.

REPRODUCIBILITY STATEMENT

Our code is provided in the anonymous link with detailed instructions on how to execute it. Our experiments only use benchmark datasets, which are publicly available. The hyperparameter details of our experiment are also provided in the appendix.

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

# A  ADDITIONAL DETAILS FOR EXPERIMENTAL SETTINGS

## A.1  DATASETS

**CIFAR-10.** The CIFAR-10 (Krizhevsky et al., 2009) contains 50,000 training images and 10,000 testing images with the size of $3 \times 32 \times 32$ in 10 classes.

**GTSRB.** The GTSRB (Stallkamp et al., 2012) contains 39,209 training images and 12,630 testing images in 43 classes. In our experiments, the images are resized to $3 \times 32 \times 32$.

**ImageNet200.** ImageNet (Deng et al., 2009) contains over 1.2 million high-resolution images in 1,000 classes. In our experiments, we randomly select 200 classes from the ImageNet dataset as our ImageNet200 dataset. Each class has 1300 training images and 50 testing images. The ImageNet images are resized to $3 \times 224 \times 224$.

## A.2  ATTACK THREAT MODEL

Table 7: Threat model across attack methods in the baselines. • indicates that the attack satisfies the property; ∘ indicates that the attack does not satisfy the property.

|  | BadNets | Blend | WaNet | IAD | AdvDoor | Bpp | LC | Narcissus | SSDT | Adap-blend | Grond |
|---|---|---|---|---|---|---|---|---|---|---|---|
| Clean-Label | ∘ | ∘ | ∘ | ∘ | ∘ | ∘ | • | • | ∘ | ∘ | • |
| No Control over Training | • | • | • | ∘ | • | ∘ | • | • | ∘ | • | ∘ |
| Invisible Trigger | ∘ | ∘ | • | ∘ | • | • | ∘ | ∘ | ∘ | ∘ | • |

## A.3  BACKDOOR ATTACKS

Our attack is compared with ten well-known and representative attacks: BadNets (Gu et al., 2019), Blend (Chen et al., 2017), WaNet (Nguyen & Tran, 2021), IAD (Nguyen & Tran, 2020), Adv-Door (Zhang et al., 2021), BppAttack (Wang et al., 2022b), LC (Turner et al., 2019), Narcissus (Zeng et al., 2023), Adap-Blend (Qi et al., 2023), and SSDT (Mo et al., 2024).

Table 8: The backdoor training settings.

| Config | Value |
|---|---|
| Optimizer | SGD (other architectures), AdamW (InceptionNeXt) |
| Weight decay | $5 \times 10^{-4}$ |
| learning rate | 0.01 |
| epoch | 200 (GTSRB, CIFAR10), 100 (ImageNet200) |
| learning rate schedule | MultiStepLR (100, 150), CosineAnnealingLR (ImageNet200) |
| poison rate | 0.05 |
| BadNets trigger | $3 \times 3$ |
| Blend trigger | random Gaussian noise and blend ratio 0.2 |
| Adap-blend trigger | "hellokitty_32.png" and blend ratio of 0.2 |
| Narcissus trigger size | $\epsilon = 16$ for both inference and training |

Like Narcissus, our attack uses the class bird (CIFAR10) as the target class. For ImageNet200, we use the stingray as the target. The Grond poisoning rate (ImageNet200) used for results in Table 5 is 0.5%. For GTSRB, we use the speed limit (50) as the target. The Grond poisoning rate (GTSRB) used for results in Table 5 is 1.74%. AdvDoor uses the same trigger and target class as ours. More details are provided in Table 8. For other attacks and hyperparameters not mentioned, we use the default setting from the original papers or open-source implementations.

## A.4  BACKDOOR DEFENSES

We evaluate our attack and baseline attacks against 16 defenses, including **4 pruning-based methods** (FP (Liu et al., 2018a), ANP (Wu & Wang, 2021), CLP (Zheng et al., 2022) and RNP (Li et al., 2023a)), **5 fine-tuning-based methods** (vanilla FT, FT-SAM (Zhu et al., 2023), I-BAU (Zeng et al., 2022), FST (Min et al., 2023) and BTI-DBF(U) (Xu et al., 2024b)), **5 backdoor model detections** (NC (Wang et al., 2019), Tabor (Guo et al., 2020), FeatureRE (Wang et al., 2022a), Unicorn (Wang et al., 2023) and BTI-DBF (Xu et al., 2024b)), and **2 backdoor input detections** (Scale-up (Guo et al., 2023) and IBD-PSC (Hou et al., 2024)).

**ANP, CLP, RNP, FST, BTI-DBF, BTI-DBF(U), FeatureRE, Unicorn**. We use the implementation and default hyperparameters from their open-source code.

**FP, vanilla FT, FT-SAM, I-BAU**. We use the implementation and default hyperparameters on BackdoorBench (Wu et al., 2022). For FT-SAM on ImageNet200, the default setting will decrease benign accuracy to 0.465, so we reduce its training scheduler to 25 epochs. Please note that the experiments on CIFAR10 with FT-SAM usually converge within 20 epochs. Therefore, decreasing the training scheduler is not harmful to the defense performance.

**NC and Tabor.** We use the implementation from TrojanZoo (Pang et al., 2022). 1% training set and 100 epoch are used for trigger inversion.

**Scale-up, IBD-PSC**. We use the implementation and default hyperparameters from BackdoorBox (Li et al., 2023b).

### A.5 HYPERPARAMETERS FOR THE INVERSED BACKDOOR FEATURE LOSS

Following the settings in BTI-DBF (Xu et al., 2024b) and BAN (Xu et al., 2024a), we use Adam and the learning rate of 0.01 to search for 20 epochs for the feature mask in Equation 4. The optimization of the mask uses 1% of training data. The $\lambda$ is 0.72. The elements in the mask are limited to continuous values between 0 and 1.

### A.6 HYPERPARAMETERS FOR TRAINING SURROGATE MODELS

Table 9 shows the hyperparameters for training surrogate models to generate TUAP.

Table 9: The backdoor training settings.

| Config | Value |
|---|---|
| Optimizer | SGD (other architectures), AdamW (InceptionNeXt) |
| Weight decay | $5 \times 10^{-4}$ |
| learning rate | 0.01 (CIFAR10, GTSRB), 0.001 (ImageNet200) |
| epoch | 200 (GTSRB, CIFAR10), 100 (ImageNet200) |
| learning rate schedule | MultiStepLR (100, 150), CosineAnnealingLR (ImageNet200) |

## B ADDITIONAL EXPERIMENTS

### B.1 INPUT-SPACE DETECTION

Table 10 shows the input-space detection results using Scale-up and IBD-PSC. We report the True Positive Rate (TPR), False Positive Rate (FPR), AUC, and F1 score in Table 10. Scale-up and IBD-PSC perform well against three baseline attacks but cannot detect backdoor inputs of Grond.

Table 10: Input-space detection results.

| Attack | Scale-up | | | | IBD-PSC | | | |
|---|---|---|---|---|---|---|---|---|
| | TPR | FPR | AUC | F1 | TPR | FPR | AUC | F1 |
| BadNets | 81.93 | 32.90 | 0.7627 | 0.7524 | 100 | 7.90 | 0.9996 | 0.9606 |
| Blend | 99.32 | 38.74 | 0.8681 | 0.8275 | 100 | 0.90 | 1.00 | 0.9953 |
| Adap-Blend | 68.72 | 18.99 | 0.7621 | 0.7297 | 53.95 | 11.77 | 0.8731 | 0.6495 |
| Ours (pr=5%) | 24.40 | 17.69 | 0.5463 | 0.3409 | 0.00 | 10.33 | 0.5698 | 0.0 |
| Ours (pr=1%) | 18.39 | 17.96 | 0.4879 | 0.2656 | 0.00 | 5.82 | 0.0626 | 0.0 |
| Ours (pr=0.5%) | 7.05 | 16.19 | 0.4034 | 0.1113 | 0.00 | 4.82 | 0.1087 | 0.0 |

### B.2 ADVERSARIAL BACKDOOR INJECTION FOR OTHER ATTACKS

Figure 5 shows additional Adversarial Backdoor Injection results against models without defense.

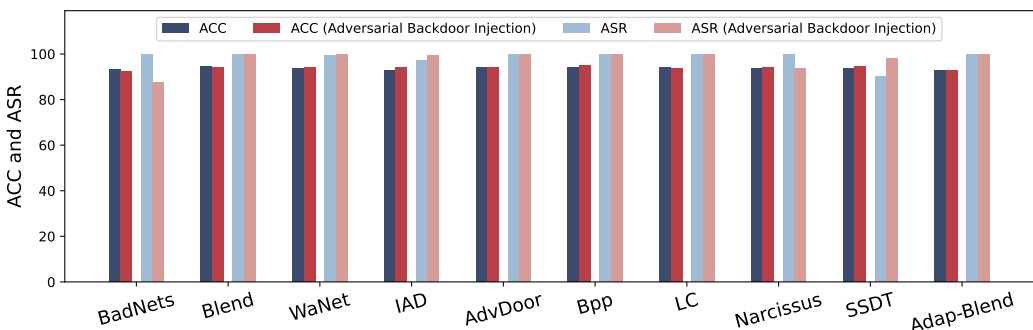

Figure 5: The attack performance (no defense) when combined with Adversarial Backdoor Injection.

## B.3 DIFFERENT ARCHITECTURES WITH DIFFERENT SURROGATE MODELS

We evaluate `Grond` with four additional victim architectures in Table 11: VGG16, DenseNet121, EfficienNet-B0, and InceptionNeXt-Tiny. In addition, as `Grond` requires a surrogate model to generate TUAP as the backdoor trigger, we provide the results when TUAP is generated using different architectures for the surrogate model. For each architecture, TUAP is generated by the same architecture or ResNet18 to perform our attack. In Table 11, we use the three most powerful defenses according to Tables 2 and 3. Regardless of the model's architectures or the architectures for TUAP, `Grond` bypasses most defenses. This is because the TUAP contains semantic information of the target class and can be transferred among different architectures (Moosavi-Dezfooli et al., 2017). In a few cases, using TUAP generated by the same architecture shows better attack performance. For example, conducting `Grond` on InceptionNeXt-Tiny with TUAP generated by InceptionNeXt-Tiny shows ASRs above 90%, but also a much lower ASR when using TUAP generated by ResNet18. We conjecture that transferring TUAP from ResNet18 to InceptionNeXt-Tiny is more difficult than transferring it to other architectures due to the large convolution kernel design of InceptionNeXt.

Table 11: `Grond`'s performance against defenses using different architectures with CIFAR10. The poisoning rate is 5%. The surrogate indicates the architecture used to generate TUAP as the trigger.

| Victim | Surrogate | No Defense | | FT-SAM | | I-BAU | | FST | |
|---|---|---|---|---|---|---|---|---|---|
| | | BA | ASR | BA | ASR | BA | ASR | BA | ASR |
| VGG16 | ResNet18 | 92.69 | 95.31 | 92.72 | 78.42 | 90.10 | 14.53 | 89.12 | 92.68 |
| | VGG16 | 92.57 | 90.10 | 92.22 | 95.14 | 90.20 | 76.51 | 91.72 | 90.58 |
| DenseNet121 | ResNet18 | 92.39 | 95.62 | 90.98 | 23.88 | 86.73 | 48.14 | 90.77 | 88.94 |
| | DenseNet121 | 92.38 | 81.07 | 91.10 | 16.91 | 90.90 | 54.76 | 91.13 | 71.29 |
| EfficienNet-B0 | ResNet18 | 87.7 | 96.23 | 84.05 | 71.07 | 87.64 | 95.41 | 82.07 | 97.67 |
| | EfficienNet-B0 | 86.92 | 92.61 | 83.77 | 71.17 | 86.93 | 92.13 | 82.45 | 68.72 |
| InceptionNeXt-Tiny | ResNet18 | 85.07 | 91.83 | 85.07 | 2.17 | 85.25 | 91.67 | 82.78 | 3.82 |
| | InceptionNeXt-Tiny | 85.54 | 96.24 | 85.64 | 90.14 | 85.49 | 97.21 | 83.92 | 97.29 |

## B.4 CACULATING TAC VALUES

To study the backdoor neurons and their effects, we calculate the TAC (Zheng et al., 2022) values with knowledge of backdoor triggers. The TAC value measures the difference when the network accepts benign and backdoor inputs. A large TAC value means the corresponding neuron is strongly related to backdoor behaviors. Specifically, TAC is defined as:

$$\text{TAC}_l^{(k)}(\mathcal{D}_c) = \frac{1}{|\mathcal{D}_c|} \sum_{\boldsymbol{x} \in \mathcal{D}_c} ||f_l^{(k)}(\boldsymbol{x}) - f_l^{(k)}(G_x(\boldsymbol{x}))||_2, \qquad (5)$$

where $f_l^{(k)}$ is the $k_{th}$ channel of the $l_{th}$ layer. $\mathcal{D}_c$ consists of a few benign samples. Note that TAC is only used for analyzing the backdoor behaviors, and it cannot be used for defense, as it requires access to backdoor triggers, which is not realistic.

## B.5 TAC PLOTS

In Figure 6, we show the TAC plots of other attacks as a supplementary of Figure 4. Other attacks also show a part of the prominent neurons with significantly higher TAC values than other neurons. For clearer demonstration, we also provide sorted TAC value plots in Figure 7, which sorts the TAC values in Figures 4 and 6. Figure 7 clearly demonstrates the existence of prominent neurons, and `Grond` is more stealthy.

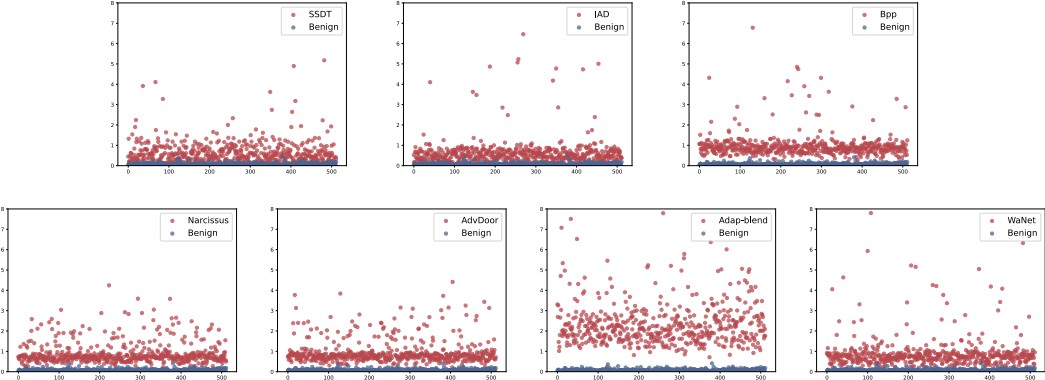

Figure 6: TAC plots of other attacks.

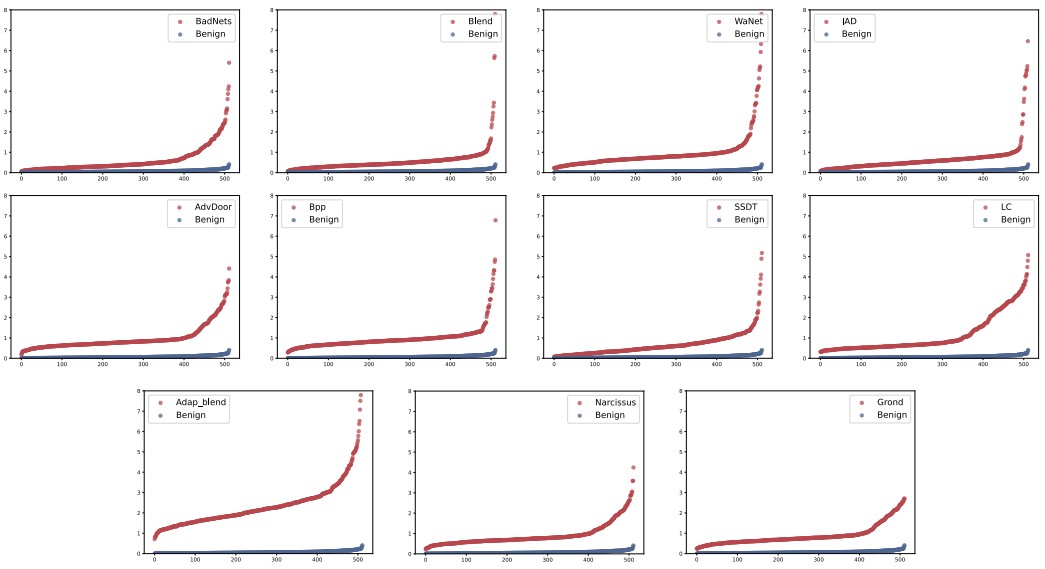

Figure 7: TAC plots of sorted TAC values, which show the prominent neurons of baseline attacks. However, such prominent neurons are not found in our attack.

## B.6 EXAMPLES OF POISONED IMAGES

Figure 8 shows four training images from ImageNet200 when applied with TUAP. Please note the images are only meant to demonstrate both trigger and poisoning perturbations invisibility (i.e., clean-label). In our experiments, we only poison training images from the class "stingray" to inject the backdoor. The first row depicts poisoned images, while the second contains clean ones. Finally, the third row contains the residual of the first two rows. Notice that `Grond` does not introduce any visible difference to the clean images.

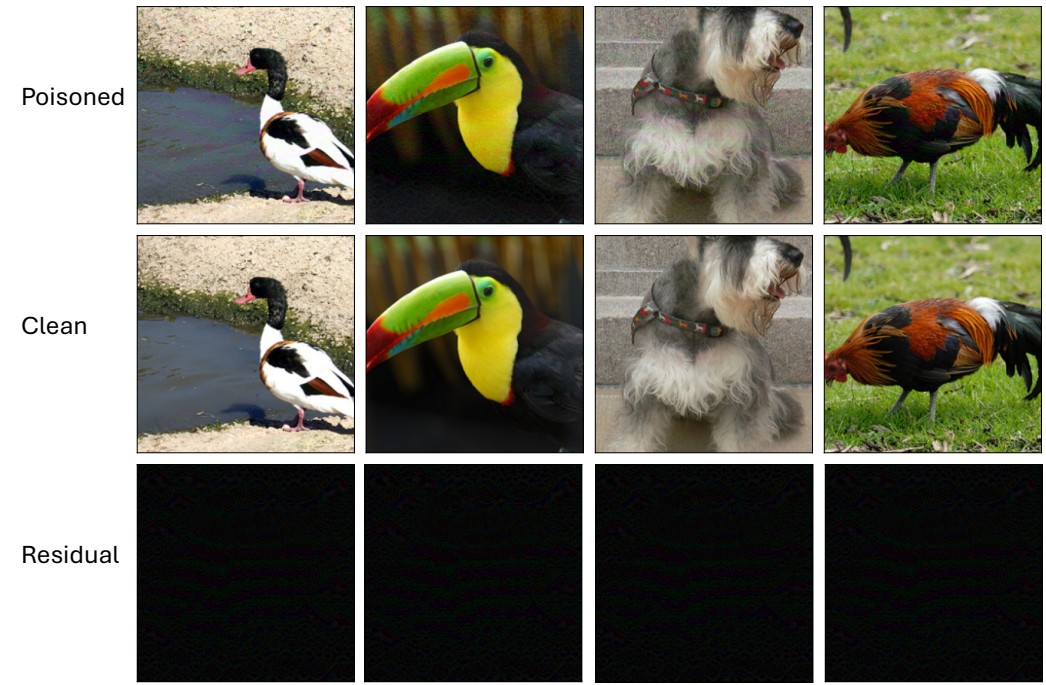

Figure 8: Examples of poisoned ImageNet200 images by `Grond`. We only poison training images from the class "stingray" in our experiments with ImageNet200.

## B.7 EXAMPLES OF GRAD-CAM IMAGES AND FEATURE VISUALIZATION

Figure 9 shows the heatmap using Grad-CAM Selvaraju et al. (2017), highlighting the important areas in the images that contribute to the prediction. It is clear that the clean input and poisoned input use similar image pixels for the model to do the classification.

Figure 10 shows the latent feature from a Grond backdoor model in 2-D space by t-SNE Van der Maaten & Hinton (2008).

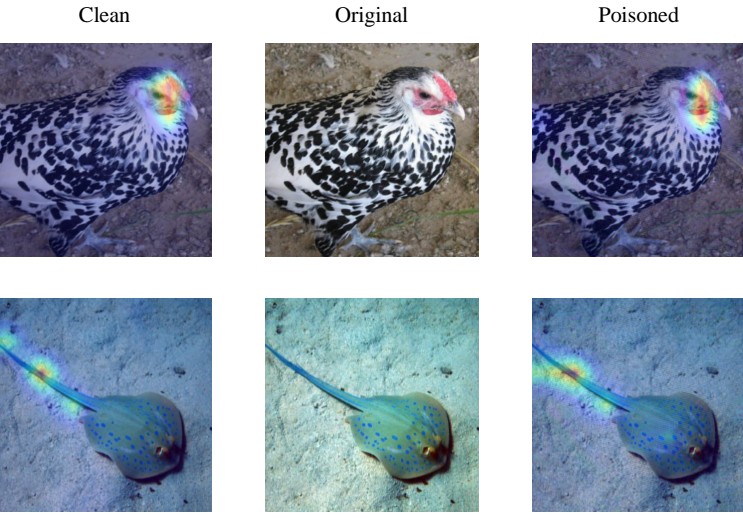

Figure 9: Examples of Grad-CAM heatmap with ImageNet200 images by the `Grond` model.

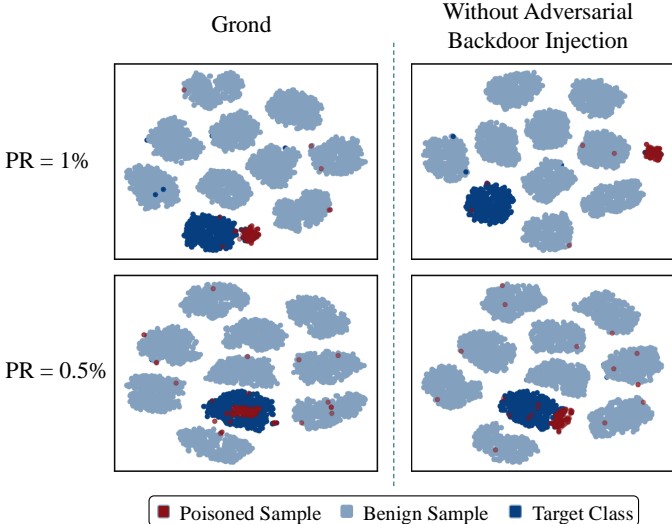

Figure 10: Examples of feature visualization of `Grond` and without our adversarial backdoor injection.

## C  OTHER LIMITATIONS

**Dirty-label `Grond`.** `Grond` is a clean-label backdoor attack that uses an invisible trigger and does not change the original labels of the poisoned samples. Since in our threat model, the attacker, having access to the training data, provides a poisoned model to the user, we could also explore the effect of a dirty-label backdoor attack to insert an all-to-one backdoor. Theoretically, a dirty-label attack could simplify the backdoor insertion and require fewer poisoned samples, which could potentially reduce our attack's overhead and improve its performance.

