# OpenReview forum: "Grond: A Stealthy Backdoor Attack in Model Parameter Space"
_ICLR.cc/2025/Conference — ICLR 2025 Conference Withdrawn Submission_

### Official Review · Reviewer_Q4Xe · 2024-10-30

**Soundness:** 3
**Presentation:** 3
**Contribution:** 2
**Rating:** 5
**Confidence:** 4

**Summary:**

In this paper, the researchers propose a new backdoor attack scheme to combat existing defense strategies based on model repairing. The core idea of this scheme is very simple and easy to understand. Specifically, this scheme first generates a trigger using TUAP, and then uses this trigger to poison the model. During the process of implanting the backdoor, it modifies parameters with higher activation values, thereby enhancing the stealth of the backdoor attack in the parameter space. Additionally, experimental results demonstrate the effectiveness of this scheme. However, both trigger generation and adversarial backdoor injection are based on existing works, so the innovation of this research is limited.

**Strengths:**

The experimental results of this approach are convincing. The results indicate that the approach can bypass existing defense mechanisms while maintaining a high attack success rate.

**Weaknesses:**

This work lacks novelty because two key steps—trigger generation and adversarial backdoor injection—are based on existing studies [1], [2].

**Questions:**

1. The experimental section should clearly specify the number of clean samples used in the pruning-based and fine-tuning-based approaches, which is not clearly stated in the paper.
2. The authors list some ImageNet200 backdoor samples; however, despite the authors claiming that this backdoor attack is stealthy in input space, feature space, and parameter space, a close examination of these backdoor samples reveals obvious perturbations. The authors should evaluate the quality of the backdoor images, especially in comparison with some imperceptible backdoor attacks, such as [3].
3. It would be more convincing if the authors could provide Grad-CAM images of the backdoor samples and visualise their distribution in feature space.


[1] Moosavi-Dezfooli, Seyed-Mohsen, et al. "Universal adversarial perturbations." Proceedings of the IEEE conference on computer vision and pattern recognition. 2017.
[2] Zheng, Runkai, et al. "Data-free backdoor removal based on channel lipschitzness." European Conference on Computer Vision. Cham: Springer Nature Switzerland, 2022.
[3] Doan, Khoa, et al. "Lira: Learnable, imperceptible and robust backdoor attacks." Proceedings of the IEEE/CVF international conference on computer vision. 2021.

---

> ### Author Response · Authors · 2024-11-22
>
> > The number of clean samples used in the pruning-based and fine-tuning-based approaches
>
> Following the default setting in ANP, RNP, etc, we use 1\% of the clean training data for defenses.
>
> > A close examination of these backdoor samples reveals obvious perturbations.
>
> We agree that perturbations are perceptible when inspecting closely.
> We follow the common practice in the backdoor and adversarial example research, where $L_\inf  = 8$ is used as a proxy to represent imperceptibility.
>
>
> > Grad-CAM images of the backdoor samples and visualize their distribution in feature space.
>
> We have updated the draft with the Grad-CAM and t-SNE plot results in Appendix B.7 in the updated PDF draft. It is clear that the clean input and poisoned input use similar image pixels for the model to do the classification.

---

### Official Review · Reviewer_xjYn · 2024-10-31

**Soundness:** 3
**Presentation:** 3
**Contribution:** 2
**Rating:** 3
**Confidence:** 4

**Summary:**

This paper presents Grond, a backdoor attack that achieves enhanced stealth across input, feature, and parameter spaces to avoid detection. Through Adversarial Backdoor Injection, Grond disperses backdoor effects across multiple neurons, making it harder to identify using parameter-space defenses. Extensive experiments on datasets like CIFAR-10, GTSRB, and ImageNet200 show that Grond outperforms other attacks in evading both pruning and fine-tuning-based defenses, highlighting its robustness and adaptability.

**Strengths:**

This paper introduces Grond, a novel backdoor attack with comprehensive stealth across input, feature, and parameter spaces. Using Adversarial Backdoor Injection, Grond disperses backdoor effects across neurons, enhancing its stealth and evading parameter-space defenses. Extensive testing on multiple datasets (CIFAR-10, GTSRB, ImageNet200) and defenses demonstrates Grond's effectiveness and adaptability across diverse scenarios and model architectures.

**Weaknesses:**

1. Limited Threat Model in Terms of Defender Capabilities: The paper's threat model lacks a thorough consideration of the defender’s capabilities, particularly regarding proactive measures they could take to identify and mitigate backdoors prior to deployment. This omission may limit the applicability of the model to real-world scenarios where defenders could leverage more advanced tools and strategies.
2. Lack of Comparison with Stealthy Clean-Label Backdoor Attacks: The paper does not include a comparison with other existing stealthy clean-label backdoor attacks, such as Hidden Trigger Backdoor Attacks (HTBA).
3. Limited Range of Defense Methods Evaluated: The paper tests Grond against a small selection of defense methods, primarily focusing on pruning and fine-tuning-based defenses.

**Questions:**

Can this type of backdoor attack be detected by sample-based detection defenses?

---

> ### Author Response · Authors · 2024-11-22
>
> > Limited Threat Model
>
> Please see the general reply on the threat model.
>
> > Lack of Comparison with Stealthy Clean-Label Backdoor Attacks
>
> We have included two representative stealthy clean-label backdoor attacks, namely Narcissus (SOTA attack) and label-consistent backdoor.
> Like HTBA, these two backdoor attacks are clean-label, i.e., stealthy in input space.
> We believe that our experiments on Narcissus and the label-consistent backdoor could provide enough evidence to support our contributions.
>
> > Limited Range of Defense Methods Evaluated
>
> We have included all types of defenses after the backdoor training, including detection and mitigation.
> In particular, the detection includes model detection and input detection. The mitigation includes fine-tuning and pruning-based methods to remove the backdoor.
> We have included all representative defenses of both types (see Tables 2, 3, 4, and 10), based on which we believe that we can make a solid conclusion.

---

> > ### Comment · Reviewer_xjYn · 2024-11-25
> > **Reviewer Feedback**
> >
> > Since the attack takes stealthiness into consideration, I believe it is important for you to evaluate the effectiveness of sample-level defense mechanisms as well. For example, approaches like "Towards A Proactive ML Approach for Detecting Backdoor Poison Samples" and similar sample detection methods could be incorporated into your evaluation. This would provide a more comprehensive analysis, rather than focusing solely on model-level defenses such as pruning-based detection.

---

> > > ### Author Response · Authors · 2024-11-25
> > >
> > > Thank you for your advice, but we have included 2 latest sample-level (backdoor input detections) defenses, Scale-up[A] and IBD-PSC [B], in our results in Table 10.
> > > In Table 10, Grond inputs cannot be detected by these SOTA methods, but baseline attacks's inputs are detectable.
> > > We will also consider the method the reviewer advised.
> > >
> > >
> > > [A] Scale-up: An efficient black-box input-level backdoor detection via analyzing scaled prediction consistency. ICLR 2023
> > >
> > > [B] IBD-PSC: Input-level Backdoor Detection via Parameter-oriented Scaling Consistency. ICML 2024

---

> ### Comment · Reviewer_xjYn · 2024-11-25
> **Reviewer Feedback**
>
> Some articles focus on proactive defenses against backdoor attacks, while others focus on negative defenses. It is important to clearly differentiate between these approaches in the paper. Typically, once a backdoor is injected, the defenses employed are negative, such as the fine-tuning-based defenses evaluated in the main body of this paper. However, in your discussion, particularly regarding input defenses, it is crucial to address how existing proactive defenses might counter your proposed attack.
>
> Additionally, the threat model assumes overly strong conditions, which make it challenging to apply in real-world scenarios. Furthermore, the threat model mentions that the model can be provided to users, but it does not account for the possibility that users might fine-tune the model themselves. This could potentially impact the effectiveness of the backdoor attack, yet this aspect is not addressed in the paper.
>
> Given these concerns, I maintain my previous decision.

---

### Official Review · Reviewer_SomH · 2024-11-03

**Soundness:** 3
**Presentation:** 3
**Contribution:** 2
**Rating:** 3
**Confidence:** 5

**Summary:**

The paper proposes a new clean-label backdoor attack that achieves stealthiness in both the input space and the parameter space. Specifically, to achieve the stealthiness in the input space, the paper utilizes the targeted universal adversarial perturbation as the backdoor trigger. For the parameter-space stealthiness, the paper restricts the magnitude of model weight parameters by setting particularly large weights to the mean value of the corresponding layer. The evaluation is conducted on three standard benchmarks. Comparing to existing backdoor attacks, the proposed attack is more resilient to existing defense and detection methods.

**Strengths:**

1. The studied topic is important as backdoor attacks can exploit the integrity of deployed deep learning models and cause unexpected consequences.
2. The paper is overall easy to understand.

**Weaknesses:**

1. The proposed attack utilizes the targeted universal adversarial perturbation as the backdoor trigger, which is the same as an existing work [1]. To increase the stealthiness of the attack in the parameter space, the paper uses backdoor defenses to help reduce backdoor-related neurons. This technique has already been proposed in the literature [2]. The proposed attack is just a collection of existing techniques. The novelty is very limited.
2. The paper assumes that the attacker has white-box access to the training processes, meaning that the attacker has whole control over the training. And yet, the paper chooses a clean-label attack, which is quite strange. The introduction of clean-label attacks is to simulate the scenario where adversaries have no control over the labeling and training procedures. The attacker can only modify a subset of the training images, making it a realistic threat model. Since this paper assumes white-box access to the training processes, there is no need to use the clean-label setting. Can the authors explain why such a setting is needed for a successful attack?
3. According to Figure 2, the mask loss for the proposed backdoor attack is much lower than benign cases. Cannot one design a defense method by measuring the outliers of the mask loss? Clearly the mask loss for the proposed attack is much smaller, which can be easily detected.
4. Following the above point, there is no evaluation on adaptive defenses, where the defender has the knowledge of the proposed attack. Since the proposed attack uses a small-size trigger with epsilon equal to 8, simple defenses can be existing adversarial detection methods and/or (universal) adversarial training. Another defense approach could be randomly perturbing the weight parameters. As the proposed attack reduces backdoor weights, the backdoor effect may be quite brittle when weights are perturbed.



[1] Zeng, Yi, et al. "Narcissus: A practical clean-label backdoor attack with limited information." Proceedings of the 2023 ACM SIGSAC Conference on Computer and Communications Security. 2023.

[2] Cheng, Siyuan, et al. "Deep feature space trojan attack of neural networks by controlled detoxification." Proceedings of the AAAI Conference on Artificial Intelligence. Vol. 35. No. 2. 2021.

**Questions:**

Please see above comments.

---

> ### Author Response · Authors · 2024-11-22
>
> > The novelty is very limited.
>
> Please see the general reply on our contributions.
>
> In addition, we also provide results of Grond with different triggers rather than TUAP, as shown in the following table:
>
> |Attack| BA (No def.) | ASR (No def.) | BA (CLP) | ASR (CLP) | BA (FT-SAM) | ASR (FT-SAM) |
> | ---- | ---- | ---- | ---- | ---- | ---- | ---- |
> | Grond (random noise) | 94.24 | 1.28 | 94.13 | 0.97 | 93.90 | 1.84 |
> | Grond (pgd noise) | 94.77 | 69.33 | 92.57 | 46.63 | 92.40 | 24.56 |
> | Grond | 93.43 | 98.04 | 93.29 | 87.89 | 92.02 | 80.07 |
>
> We notice that random noise is not effective at all, while PGD noise is relatively more effective but is still worse than TUAP.
> We will include the experimental results in the revision.
>
>
> > Threat model.
>
> Please see the general reply on the threat model.
>
> > Can the authors explain why such a setting (clean-label) is needed for a successful attack?
>
> We agree with you that a clean label is not a must for a successful attack.
> In fact, in the clean-label attack scenario, adversaries follow a more strict threat model, and they can only modify training images in a limited way.
> We anticipate that dirty-label attacks that allow adversaries to modify training more drastically would make the attack more effective, and we leave this exploration to future work.
>
> > Cannot one design a defense method by measuring the outliers of the mask loss?
>
> This is not easy since benign models also have low backdoor mask loss.
> So, it's difficult to distinguish between benign and backdoor models based on the mask loss.
> We will clarify this point in the revised version.
>
> > Adaptive defenses
>
> Please see the general reply on the adaptive defense.

---

> > ### Comment · Reviewer_SomH · 2024-11-25
> >
> > Thanks for the response. However, the rebuttal does not sufficiently address my concerns. I will keep my score.

---

### Official Review · Reviewer_tLTG · 2024-11-04

**Soundness:** 3
**Presentation:** 3
**Contribution:** 1
**Rating:** 3
**Confidence:** 5

**Summary:**

The authors propose a backdoor attack against image classification models that improves robustness and stealthiness. Current backdoor attacks are "easily spotted by examining the parameter space". The authors propose an Adversarial Backdoor Injection method that prunes weights of the backdoored network after each training epoch whenever they deviate too strongly from the mean weight within each layer. The authors evaluate their attack on relatively small-scale image datasets, such as CIFAR-10 and a 200-class subset of ImageNet, which includes nine backdoor removal and seven backdoor detection methods. The results show their attack is robust and undetectable against all surveyed defences.

**Strengths:**

- Effectiveness: The paper's results are promising and show improvement over other attacks in all dimensions.
- Ablation studies: The authors conducted extensive experiments across multiple datasets (CIFAR-10, GTSRB, ImageNet200) and architectures to analyse their attack’s effectiveness.
- Presentation: The methodology and results are presented clearly, making the paper easy to follow.

**Weaknesses:**

- Lack of Novelty: The approach of pruning weights to enhance stealth is not particularly original and provides only limited new insights into defending against these types of attacks. This limits the novelty of the proposed method.

- Assumption of a Strong Attacker: The paper assumes a white-box threat model with complete control over the training process. This setting, also known as a ‘supply chain attack’ [A], is extremely challenging (hopeless?) to defend against. It may not represent more realistic attacks or limited-access scenarios. This was stated in previous works [A] and others even show that provably undetectable backdoors can be implanted into models in this setting [B].

- Lack of theoretical insights: There are no clear reasons why Grond should perform better than existing attacks, and the authors do not provide insights on the 'why' question.

- Lack of adaptive defenders: From a security perspective, it appears that Grond does not evaluate an adaptive defender who knows the attack strategy used by Grond. For instance, pruning weights in the way the authors proposed could make the attack detectable.

-------
[A] Hong, Sanghyun, Nicholas Carlini, and Alexey Kurakin. "Handcrafted backdoors in deep neural networks." Advances in Neural Information Processing Systems 35 (2022): 8068-8080.

[B] Goldwasser, Shafi, et al. "Planting undetectable backdoors in machine learning models." 2022 IEEE 63rd Annual Symposium on Foundations of Computer Science (FOCS). IEEE, 2022.

**Questions:**

- Why does Grond work better than any other method?

- Why is the white-box setting significant?

- Do adaptive defenders exist that can detect or remove Grond?

---

> ### Author Response · Authors · 2024-11-22
>
> > Why does Grond work better than any other method?
>
> The design of Grond is aware of the existence of parameter-space defenses, while other backdoor attacks do not consider parameter-space defenses.
> So, it's anticipated that Grond outperforms other backdoor attacks against parameter-space defenses.
> In addition, we also provide analyses of feature space (Figure 2) and parameter space (Figure 4) to show the effectiveness of Grond.
> Compared to all other baselines in our experiments, Grond shows better stealthiness in feature space and parameters space.
> All other baseline attacks show a set of prominent neurons with much higher TAC values than other neurons.
> In our TAC pruning experiment (Figure 4), we show that removing these neurons with higher TAC values could effectively mitigate backdoor attacks.
> However, as Grond constrains the parameter while training and spreads the backdoor effect to more neurons, the backdoor neurons's TAC values are close to benign neurons.
> Pruning these neurons will significantly reduce benign accuracy.
>
> > Why is the white-box setting significant?
>
> Please see the general reply on the threat model.
>
> > Do adaptive defenders exist that can detect or remove Grond?
>
> Please see the general reply on the adaptive defense.
>
> > Lack of theoretical insights
>
> We did not provide a theoretical analysis since the main contribution of our paper is to recognize the problem that current types of backdoor attacks can be mitigated by parameter-space defenses, and we provide a generalized solution (see Section 4.4) to tackle this problem. We leave theoretical analysis to future work.

---

> ### Author Response · Authors · 2024-11-22
>
> > undetectable backdoors
>
> The undetectable concept in [B] mainly focuses on black box-undetectable backdoors, where the defender has no access to the model weight, architecture, etc.
> For white box-undetectable backdoors, [B] is only applicable on Fourier feature networks with no ReLU activation layers (a sigmoid activation at the end will preserve the backdoor) and on fully connected networks with just 1 hidden layer with a ReLU activation, based on which we can conclude that this backdoor is not applicable in practice. Fourier feature networks usually have at least 1 hidden layer (usually 3 or 4), and the same for fully connected networks. This further confirms that these types of backdoors are not applicable in practice.
>
> For the undetectability under white-box access, [C] investigates the existence of backdoor attacks to obfuscated neural networks, which are undetectable even when given white-box access. [C] goes further by combining ideas from steganography to inspire backdoor schemes in large language models.
>
> However, both [B] and [C] only lay solid theoretical foundations, but it is still an open question of how to build practical instantiations based on these theoretical constructions.
>
> [B] Planting undetectable backdoors in machine learning models. FOCS 2022.
>
> [C] Injecting Undetectable Backdoors in Obfuscated Neural Networks and Language Models. to appear at NeurIPS 2024.

---

> > ### Comment · Reviewer_tLTG · 2024-11-26
> >
> > Thank you for your response.
> >
> > > "The attacker has white-box access to the training processes, the training data, and the model weights. During training, poisoned images do not contain visible patterns for human inspectors, so the labels of poisoned images are the same as the images’ original class, i.e., clean-label."
> >
> > If the attacker has white-box access to the entire training procedure, they can also modify the images arbitrarily. The restriction in your threat model does not make sense to me.  It also appears that your threat model is missing the most important part - the defender's capabilities and goals.
> >
> > My issue is that paper (1) lacks relevancy, as it only looks at relatively small models and datasets that provide limited insights into today's problems. It would be acceptable if the paper offered theoretical insights, but it did not. (2) The paper lacks novelty as it proposes an attack in a setting where many attacks already exist that are likely difficult (impossible?) to defend against with meaningful preservation of model accuracy. (3) Its claims are questionable as experiments on adaptive defenders or detection algorithms are not included in the paper. The authors themselves acknowledge that they "believe that future white-box adaptive defenses may mitigate Grond". For these reasons, I will keep my current score.

---

### Author Response · Authors · 2024-11-22

Dear reviewers and ACs,

Thank you for your careful consideration.
We are glad to see that all reviewers are happy with the presentation and soundness of our work, where they see that our experiments are extensive (reviewers tLTG, xjYn) and convincing (reviewer Q4Xe).
We are also happy to see that all of our contributions are recognized by reviewers. In particular, the topic of (parameter-space) backdoor defense is important (reviewer SomH), and our attack is effective against model-space mitigation (reviewer tLTG, xjYn, Q4Xe).
However, we want to emphasize the most important contribution of this work is that current types of backdoor attacks can be mitigated by parameter-space defenses, as this observation is important to future research in the backdoor community.
We will also point out factual errors from reviews about the threat model of this work, the challenges of supply-chain attacks, and adaptive defenses.
Taking the suggestions of the reviewers, new experiments on state-of-the-art supply-chain backdoor attacks are also provided.

We would like to clarify the main contribution and novelty of this paper.
For the first time, we systematically show that current backdoor attacks, including different types of state-of-the-art backdoor attacks, are vulnerable to parameter-space backdoor defenses.
This observation is new and important for future backdoor attack and defense research, and it also has a substantial influence on real-world backdoor mitigation, indicating that parameter-space defenses should get more attention from both academia and industry.

Below, we will clarify reviewers' common concerns about the threat model, supply-chain attacks, and adaptive defenses.

**Generalization of our threat model.**
We agree with reviewers that in our threat model, adversaries can control the training of the backdoored model, which is a strong assumption and similar to supply-chain backdoor attacks.
However, we disagree that this threat model is too limited to be studied.
In particular, our threat model follows the common practice in established backdoor literature [1, 2, 3, 4, 5, 6, 7, 8, 9, 10, 11, 12, 13], where the backdoored model is delivered as a product to the victim.
Our analysis further advances the practical application and understanding of the related research.

**Supply-chain attacks.**
We thank reviewers for pointing out the supply-chain backdoor attacks that were not considered in the draft and have the same threat model as Grond.
However, we disagree that supply-chain attacks are extremely challenging to defend against.
In particular, we ran additional experiments with recent supply-chain backdoor attacks, showing that they are vulnerable to parameter-space defenses in the following table,


|Attack| BA (No def.) | ASR (No def.) | BA (CLP) | ASR (CLP) | BA (FT-SAM) | ASR (FT-SAM) |
| ---- | ---- | ---- | ---- | ---- | ---- | ---- |
| DFST [7] | 95.23 | 100 | 92.43 | 3.53 | 94.70 | 0.00 |
| DFBA [6] | 88.99 | 100 | 88.96 | 9.57 | 86.03 | 5.24 |
| SSDT [3] | 93.70 | 90.30 | 93.66 | 1.20 | 93.15 | 0.60 |

where DFBA [6] is the state-of-the-art supply-chain attack among all 13 attacks we looked at.
In addition, DFBA is a better alternative to the Handcrafted backdoor [5] (mentioned by the reviewer tLTG), as DFBA was published in the last few weeks and directly compared with the Handcrafted backdoor and shows better performance in their paper.
Therefore, we used DFBA rather than the Handcrafted backdoor [5] in our experiment.
We will add a section discussing supply-chain attacks regarding the threat model, vulnerability to parameter-space defenses, and how Grond could possibly improve supply-chain backdoors.


**Adaptive defense.**
The adaptive defense refers to the defender knowing the design of the attack, which has not been extensively studied in backdoor research.
Our TAC pruning experiment (i.e., oracle analysis) follows a threat model in which the adversary even knows the backdoor trigger of Grond rather than just the method design.
So, our TAC analysis provides a stronger defense than regular adaptive backdoor defense design that follows adversarial example research [14, 15].
Our analysis in Section 4.5 and Figure 4 shows that Grond is much more robust to adaptive defenses than other backdoor attacks, but we believe that future white-box adaptive defenses may mitigate Grond.


We hope that our clarification can address your concerns. We look forward to hearing from you and remain at your disposal should you have any comments/suggestions.

Best regards,

Authors of Grond

---

> ### Author Response · Authors · 2024-11-22
>
> # Reference
>
> Supply-chain attack list:
> 1. Imperceptible Backdoor Attack: From Input Space to Feature Representation
>     - How the backdoor training was controlled: One additional term is used in loss to shorten the distance between benign and malicious features.
>     - Publication: IJCAI 2022
>
> 2. DEFEAT: Deep Hidden Feature Backdoor Attacks by Imperceptible Perturbation and Latent Representation Constraints
>     - How the backdoor training was controlled: The latent feature is constrained to reduce distinguishability between benign and poisoned features.
>     - Publication: CVPR 2022
>
> 1. Robust Backdoor Detection for Deep Learning via Topological Evolution Dynamics
>     - How the backdoor training was controlled: Introducing additional terms in the loss for the Source-Specific and Dynamic-Triggers attack, which obscures the difference between normal samples and malicious samples.
>     - Publication: Security and Privacy (SP) 2024
>
> 1. A Data-free Backdoor Injection Approach in Neural Networks
>     - How the backdoor training was controlled: Designing a novel loss function for fine-tuning the original model into the backdoored one using the substitute data.
>     - Publication: USENIX Security 2023
>
> 1. Handcrafted Backdoors in Deep Neural Networks
>     - How the backdoor training was controlled: Directly manipulating a model’s weights.
>     - Publication: NeurIPS 2022
>
>
> 1. Data Free Backdoor Attacks
>     - How the backdoor training was controlled: Modifying a few parameters of a classifier to inject a backdoor.
>     - Publication: NeurIPS 2024
>
>
> 1. Deep Feature Space Trojan Attack of Neural Networks by Controlled Detoxification
>     - How the backdoor training was controlled: Proposing a controlled detoxification technique (in the training process) that restrains the model from picking up simple features.
>     - Publication: AAAI 2021
>
> 1. Composite Backdoor Attack for Deep Neural Network by Mixing Existing Benign Features
>     - How the backdoor training was controlled: Using similarity loss (SIM) measures to sample representation distances to make the training more stable
>     - Publication: CCS 2020
>
> 1. Enhancing Backdoor Attacks With Multi-Level  MMD Regularization
>     - How the backdoor training was controlled: Introducing additional terms in loss to reduce the distributional differences at multi-level representations.
>     - Publication: TDSC 2022
>
> 1. Backdoor Attack with Imperceptible Input and Latent Modification
>     - How the backdoor training was controlled: Introducing a Wasserstein-based regularization in the loss for the latent representations of the clean and manipulated inputs
>     - Publication: NeurIPS 2021
>
> 1. Towards Practical Deployment-Stage Backdoor Attack on Deep Neural Networks
>     - How the backdoor training was controlled: Directly replacing a subnet of a benign model with a malicious backdoor subnet, which builds a backdoor model.
>     - Publication: CVPR 2022
>
> 1. Simtrojan: Stealthy Backdoor Attack
>     - How the backdoor training was controlled: Introducing an additional term in the loss to reduce the distance between benign and backdoor features.
>     - Publication: ICIP 2021
>
> 1. Bypassing Backdoor Detection Algorithms in Deep Learning
>     - How the backdoor training was controlled: Designing a new loss function to minimize the difference of benign and backdoor features.
>     - Publication: EuroSP 2020
>
> Other reference:
>
> 14. On Evaluating Adversarial Robustness
>     - arXiv:1902.06705
>
> 15. On Adaptive Attacks to Adversarial Example Defenses.
>     - Publication: NeurIPS 2020

---

### Author Response · Authors · 2024-12-04

# Final Summary

We thank the reviewers for their efforts.
We also want to re-stress several points that we have different opinions from the reviewers.

## Threat model and proactive defense.

Backdoor attacks with white-box access to the training process or the capability to directly modify the models' weights are generally accepted assumptions in backdoor research, especially regarding supply-chain attacks.
Backdoor models provided as a service is reasonable due to the high cost of training models from scratch.
We include a comparison with the latest supply-chain attacks (SSDT, S&P 2024) in the submission. In the rebuttal, we include two more (DFST, AAAI 2021, and DFBA, NeurIPS 2024).

In addition, as reviewer xjYn suggested, we also include a proactive sample-level detection [16], where the defender directly has access to the training process.
As shown in the following table, [16] is ineffective against Grond when the poisoning rate is lower than 5\%, with a high false positive rate and low recall.

| Attack | ACC| ASR | Recall | FPR |
|----|----|----|----|----|
|BadNets(pr=5\%)| 93.18 | 99.96 | 2500/2500 | 1568/47500 |
| Grond(pr=5\%) | 93.84 | 99.41 | 2499/2500 | 671/47500 |
|Grond(pr=2.5\%)| 93.81 | 95.83 | 115/1250 | 7220/48750 |
|Grond(pr=1\%) | 94.09 | 92.48 | 208/500 | 6690/49500 |
|Grond(pr=0.5\%)|94.36 | 92.91 | 90/250 | 6738/49750|
|Grond(pr=0.3\%)|94.22 | 90.10 | 29/150 | 6349/49850|


## The main contribution
Again, we want to emphasize that our motivation is to raise attention to using more proper evaluation baselines for backdoor attacks. Current evaluations of SOTA backdoor attacks are mainly based on input-space and feature-space defenses.
Our experiments showed that all evaluated attacks failed against at least four types of parameter-space defenses.

## Adaptive analysis
Our TAC oracle analysis is stronger than adaptive analysis because the TAC pruning directly uses the backdoor trigger to defend against attacks.
In this setting, the defender has white-box access to any possible information from the attacker.
We provided a stronger adaptive analysis based on access to the backdoor trigger, which we believe provides solid evidence justifying our contributions.

Authors of Grond

---

### Note · Authors · 2024-12-20

I have read and agree with the venue's withdrawal policy on behalf of myself and my co-authors.